# Antimicrobial Activities of Tea Polyphenol on Phytopathogens: A Review

**DOI:** 10.3390/molecules24040816

**Published:** 2019-02-25

**Authors:** Yuheng Yang, Tong Zhang

**Affiliations:** 1College of Plant Protection, Southwest University, Chongqing 400715, China; yyh023@swu.edu.cn; 2College of Resources and Environment, Southwest University, Chongqing 400715, China

**Keywords:** tea polyphenol, antimicrobial activities, phytopathogens

## Abstract

The use of natural antimicrobial compounds in crop production has gained much attention from consumers and the agricultural industry. Consequently, interest in more natural, non-synthetic antimicrobials as potential alternatives to conventional chemical pesticides to combat phytopathogens has heightened. Tea polyphenol (TP), a unique and highly important functional component of tea plants, has been reported to possess antimicrobial properties against a wide spectrum of plant pathogens. The aim of this review is to discuss the emerging findings on the mechanisms of antimicrobial action, and the antimicrobial properties of TP, including their major components, effectiveness, and synergistic effects. More studies, particularly field studies, are still necessary to establish conclusive evidence for the effectiveness of TP against phytopathogens. However, the basic conclusion from existing studies suggests that TP is a potential antimicrobial agent for pesticide reduction in agricultural systems.

## 1. Introduction

Tea (Camellia sinensis) is the most consumed beverage in the world, aside from water, and is widely believed to have positive effects on human health [1]. The Chinese have known about the medicinal benefits of green tea since around 2,737 B.C. [2]. One important aspect of green tea is the fact that it is rich in polyphenols. The major polyphenols in green tea are flavonoids. The four major flavonoids in green tea are the catechins: epicatechin (EC), epigallocatechin (EGC), epicatechin gallate (ECG) and epigallocatechin gallate (EGCG) (Figure 1) [3]. Among them, EGCG is the most abundant component of catechins in tea leaves, followed by EGC, EC and ECG [4]. Notably, EGCG is not found in other plants [5]. The usual concentration of TPs in dried green tea leaves is up to 30% by weight [6].

Numerous studies have shown that TP components have various positive effects on human health [7,8,9]. In addition, evidence shows that TP have antimicrobial effects against a variety of human pathogens [10]. However, fewer studies have examined the antimicrobial activities of TP on phytopathogens. With more awareness of the potential health risks and environmental hazards caused by chemical pesticides, the development of botanical pesticides has aroused great interest among researchers throughout the world [11]. Phenolic compounds extracted from tea have been reported to possess antimicrobial properties against a wide spectrum of plant pathogens [12]. The main objective of this review is to unify and interpret widely scattered information on antimicrobial activities of TP against phytopathogens such as fungi, bacteria and viruses derived from existing studies. We also summarize the synergistic effects of combinations of TP and other biocontrol agents. In addition, the mechanisms of antimicrobial activities of TP are discussed. We believe that the future development and application of TP as a botanical pesticide will assist in controlling phytopathogens in a less environmentally hazardous manner than existing chemical pesticides.

## 2. Antimicrobial Activities of TP on Phytopathogens

Many laboratory studies conducted over the last decades have examined that treatment of TP can inhibit the infection process of in various plant-pathogen interactions (Table 1).

### 2.1. Antifungal Activity of TP

Wang et al. tested the inhibitive effects of different TP concentrations on three species of plant pathogenic fungi, *Bipolaris maydis*, *Colletotrichum musae* and *Fusarium oxysporum* [13]. The results showed that TP significantly inhibited hyphal growth and spore germination of the three fungi, and the inhibitive effects were directly proportional to the concentration of TP solutions. Among the series concentrations, the 10 mg/mL and 5 mg/mL TP solutions showed the highest inhibitive rate against the three fungi. Of the fungi, *B. maydis* was inhibited most efficiently by TP; the 10 mg/mL and 5 mg/mL TP solution inhibited spore germination by 100% and caused protoplasm overflow and cell distortion. Subsequently, these researchers reported that TP had strong inhibition on hyphal growth and conidial germination of *Pyricularia oryzae*, which causes rice blast disease [14]. TP solutions at concentrations of 10 mg/mL and 5 mg/mL can inhibit the growth of rice blast fungi completely. Some similar results have been reported about TP antifungal activities against eight species of plant pathogenic fungi, including *Phytophthora cryptogea*, *Pestalotiopsis apiculatus*, *Colletotrichum horii*, *Sclerotinia sclerotiorum*, *C. fructicola*, *Rhizoctorzia solani*, *Lasiodiplodia theobromae*, and *F. oxysporum* [15]. The best inhibition effect was on *C. horii* with EC_50_ value of 12.84 mg/mL, followed by *P. cryptogea* with EC_50_ value of 16.01 mg/mL.

Moreover, the antifungal activity of TP has been investigated against several postharvest disease microorganisms. Liu et al. reported that TP exhibited an inhibitory effect against stem-end rot (a postharvest disease caused by *Diplodia natalensis*) in citrus fruit at concentrations of 0.1%, 0.5% and 1.0% [16]. Then, they further confirmed the inhibitory activity of TP against gray mold, another postharvest disease of grape fruits caused by *Botrytis cinerea* [17]. The results showed that the spore germination of *B. cinerea* was significantly inhibited by TP at all concentrations, and mycelium growth was significantly inhibited by TP at 0.1% and above. Yang et al. also revealed the control efficacy of TP against nectarine gray mold decay caused by *B. cinerea* [18]. The in vitro experiments showed that TP inhibited the mycelial growth in a dose-dependent manner, and the in vivo experiments showed that disease incidence and lesion diameter of gray mold of inoculated fruit were significantly lowered after being treated with TP.

The beneficial effect of TP in inhibiting obligate biotrophic fungus was also observed by Yang et al. [12], who reported on the antifungal activity of TP on *Puccinia striiformis* f. sp. *tritici* (*Pst*), an obligate biotrophic fungus that causes severe wheat stripe rust disease. The in vitro experiments showed that, at a concentration of 1.0 mg/mL, TP significantly suppressed urediniospore germination and caused the aberrant growth of germ tubes. The in vivo experiments showed that TP reduced the incidence rate and the uredia coverage rate in a dose- and application time-dependent manner. However, the ideal TP concentration range for controlling *Pst* was 20–40 mg/mL, which was much higher than the effective dosage determined in other studies. This difference is mainly attributed to the fact that TP lacks the ability to effectively enter plant tissues to inhibit the growth of the infectious hyphae, due to the absence of a good systemic activity [12].

### 2.2. Antibacterial Activity of TP

In addition to antifungal activity, TP showed inhibitory effect on various phytopathogenic bacterial infections. Fukai et al. reported the antibacterial activity of TP measured as minimum inhibitory concentration (MIC) against phytopathogenic bacteria, including eight strains of *Erwinia*, 10 strains of *Pseudomonas*, and one strain each of *Clavibacter*, *Xanthomonas* and *Agarobacterium* [19]. These bacteria tend to infect commonly cultivated vegetables such as lettuce, tomatoes, eggplants, cabbage, radish, Irish potatoes, onions, and grapes. After three days incubation of the bacterial agar plates containing different concentrations of individual TPs, EGC and EGCG showed more inhibitory effect than EC and ECG against the test bacteria, and MICs were mostly below 100 ppm. However, ECG seemed to be the least effective among them, and MICs were mostly over 1000 ppm. In addition, Alstrom conducted the in vitro and greenhouse trials to demonstrate the antibacterial activity of extracts of tea extracts against phytopathogenic bacteria (*P. syringae* pv. *pisi* race 1, *P. syringae* pv. *pisi* race 2 and *P. syringae* pv. *phaseolicola*) [20]. The antibacterial activity was measured as the diameter of the inhibition zone in agar and then by periodical viable cell counts in laboratory tests. The effect on the hypersensitive reaction and the potential for disease control after leaf infiltration and seed treatment were studied on bean plants in the greenhouse. The agar diffusion test revealed that the tea extract substantially inhibited the growth of the three pathogens (5 mm diameter well filled with tea extract), whereas the inhibition of both *P. syringae* pv. *pisi* strains on agar was not observed in periodical viable cell count test. However, in vivo tests showed that tea extract could not induce a hypersensitive reaction in broad bean leaves co-inoculated with three *P. syringae* strains. These observations indicated that these effects varied depending on bacterial strain and the method of application.

### 2.3. Antiviral Activity of TP

Having noticed the antiviral effect of tea infusion on tobacco mosaic virus (TMV) [21], Okada and Furuya tested the inhibitory effect of each TP component and its own mix against TMV and cucumber mosaic virus (CMV) on tobacco leaves [22]. The aqueous solutions of TPs were injected into the soil around the base of the plants systemically infected with TMV and CMV and showed that the number of lesions on inoculated leaves on plants treated with TPs was less than that observed on untreated plants. Further study illustrated that the number of lesions was apparently decreased with the addition of ECG and ECGC, resulting in no lesion at all at 0.5% concentration of each catechin [23].

### 2.4. Synergistic Effect of TP Combination with Certain Bioagents

Tea saponin (TS) has been shown to have significant antifungal effect against various phytopathogenic fungi [24,25,26], prompting some researchers to focus on the antifungal activity of combinations of TP and TS. Chen et al. illustrated that both TP and TS inhibited the mycelial growth of brown rot pathogen on inoculated nectarine fruit, and their combination (TP: TS, 1:2) significantly improved the controlling effect [27]. Zou et al. also confirmed the co-toxicities of TP and TS on two kinds of forest fungal pathogens, *C. horii* and *L. theobromae* [28]. These fungi cause die-back of *Eucalyptus* spp. and anthracnose of *Illicium verum* Hook. f, respectively. The synergistic effect was significant with their combination (TP: TS, 1:1) on *C. horii* and *L. theobromae*.

A previous study showed that *Hanseniaspora uvarum*, one of the most abundant yeast species isolated from the surface of grape, was an effective antagonist against *B. cinerea* [29]. The investigators further confirmed that TP at concentrations of 0.5% and 1.0% in combination with *H. uvarum* (1 × 10^6^ CFU/mL) exhibited an inhibitory effect against *B. cinerea* [17]. In addition, the synergistic effect of TP and *Candida ernobii* against postharvest disease (*D. natalensis*) was investigated [16]. The results suggested that TP at 0.1%, 0.5% and 1.0% in combination with *C. ernobii* (1 × 10^6^ CFU/mL) showed a lower infection rate of stem-end rot compared to *C. ernobii* alone. These findings suggested that the combination of TP and other biocontrol agents could effectively improve the biocontrol efficacy on phytopathogenic fungi.

## 3. Mechanisms of Antimicrobial Activity of TP

Several studies have revealed that TP shows antimicrobial effects against pathogenic organisms via several antimicrobial mechanisms (Figure 2).

### 3.1. Mechanism of Antifungal Activity of TP

Changes in cell membranes and structure affect membrane permeability, which can be indirectly detected by measuring the electrolyte leakage percentage [30]. A previous study revealed that the electrolyte leakage percentage of rice blast fungus treated with various concentrations of TP changed sharply, indicating that TP treatment changed the membrane permeability of rice blast fungus. Since the main component of cell membrane is a lipid bilayer containing a hydrophilic end and a hydrophobic end, the phenolic hydroxyl groups can bind the hydrophilic end of the lipid bilayer to agglomerate the membrane lipid, thereby destroying the cell membrane [14]. In parallel, TP induced the activities of several enzymes in inoculated plants or fruits, such as phenylalanine ammonia-lyase, catalase, peroxidase, polyphenoloxidase, chitinase and β-1, 3-glucanase [14,27]. All of these enzymes play important roles in plant defenses against fungal infection.

### 3.2. Mechanism of Antibacterial Activity of TP

The antibacterial activity of TP was also attributed to membrane perturbation. Strong bactericidal dosages of EGCG damaged the liposome membrane of *E. coli* and *S. aureus*, leading to the leakage of intramembranous materials and aggregation of the liposomes [31]. These results were in agreement with the findings that TP inhibits *Pseudomonas aeruginosa* through damage to the cell membrane with the release of small cellular molecules [32]. In addition, EGCG and ECG strongly inhibit biofilm formation of commensal and pathogenic *E. coli* strains, thereby reduce the expression of CsgD - a crucial activator of curli and cellulose biosynthesis [33].

Interestingly, some studies suggested that EGCG might directly bind peptidoglycan, a major component of the cell wall of bacteria, thereby caused the cross-linking bridges of the peptidoglycan layer to be broken, resulting in its degradation [34]. Moreover, EGCG involved into the inhibition of the lipopolysaccharide-induced TLR4 signaling and peptidoglycan-induced TLR2 signaling pathways [35,36,37]. TLRs (toll-like receptors) are a family of membrane proteins that trigger innate immune responses and are pathogen recognition proteins that have important roles in detecting microbes [38]. Therefore, it is likely that EGCG also perturb the barrier function of both cell wall and membrane possibly by disrupting the signaling pathways of bacteria. Although these findings were obtained from the pathogenic bacteria on mammals, the mechanism of antibacterial activity of TP against plant pathogens is likely to be consistent.

### 3.3. Mechanism of Antiviral Activity of TP

Unlike fungi and bacteria, TP showed different mechanisms against virus. Okada et al. identified that theaflavins, dimers of tea catechins, inhibit the multiplication of TMV by direct binding to viral RNA but not to RNA protein [39]. They presumed that the interaction of theaflavins with nucleic acids leads to the inhibition of TMV replication. This result is consistent with the inhibitory effect on human immunodeficiency virus (HIV), in which catechins inhibit HIV-1 replication by targeting several steps in HIV-1 life cycle [40]. Another study showed EGCG potently inhibited Cell-culture–derived Hepatitis C virus (HCV) entry into hepatoma cell lines as well as primary human hepatocytes and could be part of an antiviral strategy aimed at the prevention of HCV reinfection after liver transplantation [41]. These results indicated that TP showed multiple mechanisms of antiviral activity with different components.

## 4. Conclusions and Prospects

The findings mentioned in this review indicate that TP may have the potential to control various phytopathogens. In addition, TP has the characteristics of abundant resources, simple extraction process and low production cost [14]. Therefore, the employment of TP as a botanical pesticide may play a vital role in meeting the demand for organically produced plants in particular and in alleviating some environmental problems caused by the use of chemical pesticides. However, most of these conclusions were based on data from in vitro studies. To elucidate the confused inhibition mechanisms of TP, and study in vivo activity, toxicity and bioavailability are the main research directions in the future [42]. Moreover, the field utilization of TP is greatly limited by its poor permeability [12]. Recent studies reported that the encapsulation of TP using nanoemulsions improved the systemic bioavailability and efficacy [43,44,45], which may be a good way to improve TP’s permeability and the biocontrol effect for the management of phytopathogens.

## Figures and Tables

**Figure 1 molecules-24-00816-f001:**
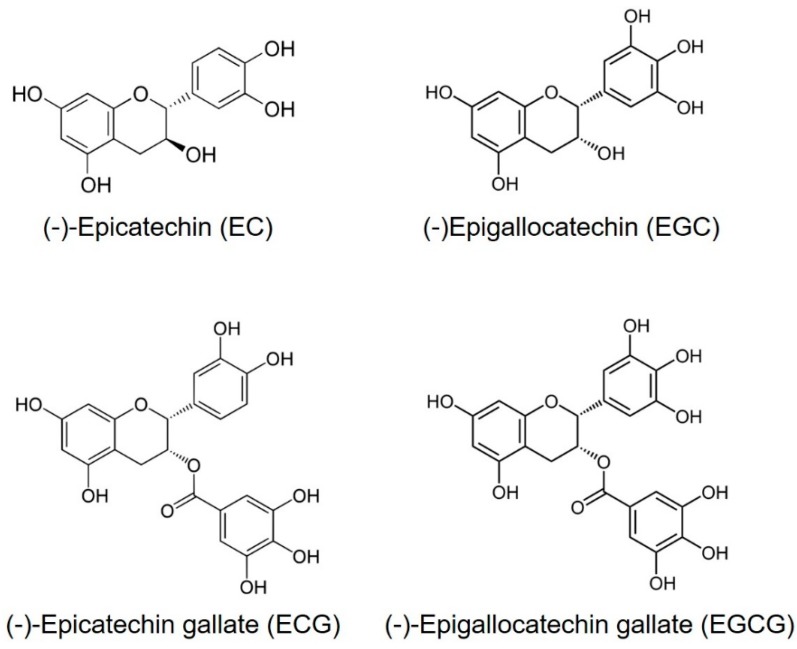
Chemical structures of major polyphenols present in green tea.

**Figure 2 molecules-24-00816-f002:**
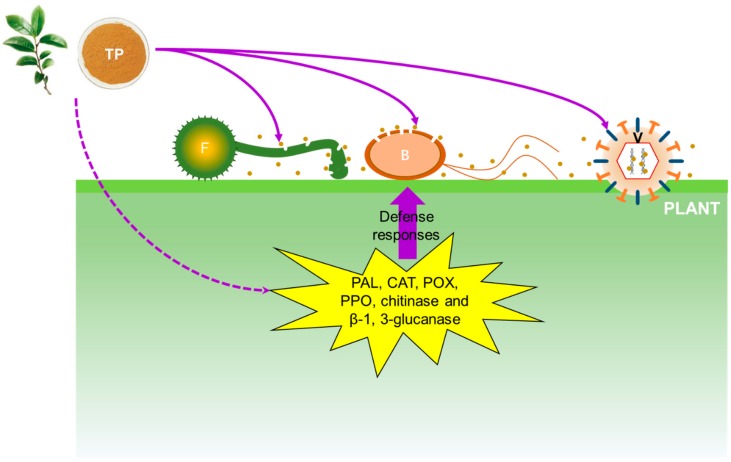
Schematic representation of the antimicrobial mechanisms of tea polyphenol. Brown dots represent tea polyphenol particles. TP: tea polyphenol; F: fungi; B: bacteria; V: virus; PAL: phenylalanine ammonia-lyase; CAT: catalase; POX: peroxidase; PPO: polyphenoloxidase.

**Table 1 molecules-24-00816-t001:** Summary of studies of tea polyphenol against phytopathogens.

Pathogen	Host	Disease	Reference
**Fungi**			
*Bipolaris maydis*	Maize	Leaf spot	[13]
*Colletotrichum musae*	Banana	Anthracnose	[13]
*Fusarium oxysporum*	Lotus/Banana	Corruption	[13,15]
*Pyricularia oryzae*	Rice	Blast	[14]
*Phytophthora cryptogea*	*Gerbera jamesonii*	Root rot	[15]
*Pestalotiopsis apiculatus*	*Camellia oleifera*	Leaf blight	[15]
*C. fructicola*	*Camellia oleifera*	Anthracnose	[15]
*C. horii*	*Illicium verum*	Anthracnose	[15,28]
*Rhizoctorzia solani*	Rice	Sheath blight	[15]
*Lasiodiplodia theobromae*	*Eucalyptus* spp.	Die-back	[15,28]
*Sclerotinia sclerotiorum*	Oilseed rape	Sclerotinia rot	[15]
*Diplodia natalensis*	Citrus fruit	Stem-end rot	[16]
*Botrytis cinerea*	Grape fruit/Nectarine fruit	Gray mold	[17,18]
*Monilinia fructicola*	Nectarine fruit	Brown rot	[27]
*Puccinia striiformis* f. sp. *tritici*	Wheat	Stripe rust	[12]
**Bacteria**			
*Erwinia* carotovora	Lettuce/Tomato/Eggplant/Cabbage/Radish/Potato/Cauliflower	Soft rot	[19]
*Clavibacter michiganensis*	Tomato	Canker	[19]
*Xanthomonas campestris*	Lettuce	Spot	[19]
*Agarobacterium tumefaciens*	Grape	Crown gall	[19]
*Pseudomonas cichorii*	Lettuce/Eggplant	Black leg	[19]
*Ps. marginalis*	Lettuce/Onion/Cabbage	Spring rot	[19]
*Ps. viridiflava*	Lettuce/Tomato	Black rot/Leaf rot	[19]
*Ps. syringae* pv. *pisi*	Bean	Halo blight	[20]
*Ps. s.* pv. *phaseolicola*	Bean	Halo blight	[20]
**Virus**			
Tobacco mosaic virus	Tobacco	Mosaic disease	[21,22,23]
Cucumber mosaic virus	Cucumber	Mosaic disease	[22,23]

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
