# Peer review of "Antimicrobial Activities of Tea Polyphenol on Phytopathogens: A Review"

_molecules, 2019, doi:10.3390/molecules24040816_

Round 1

Reviewer 1 Report

The text should be revised in order to correct the spelling mistakes.

The abbreviated species names of fungi should be corrected (e.g. B.maydis instead of „Bi. maydis“.B. cinerea instead of „Bo. cinerea“)

The following information, contained in the text of manuscript should be revised according to the source publications:

1. The activity of TP on Pestalotiopsis apiculatus and C. fructicola as a disease agents of Tea-tree (Or „Tea-oil tree“); in the Table 1 (source publication No 15)

Note: The tea-tree products can inhibit agents of leaf blight and anthracose, rather than the tea-tree would suffer of those diseases.

2. The text „A previous study revealed that the electrolyte leakage percentage of rice blast fungus treated with various concentrations of TP changed sharply, indicating that TP treatment changed the membrane permeability of rice.“ (source publication Nr. 14)– probably of rice blast fungus membrane permeability, not the "membrane permeability of rice" itself

3. The source publication No. 31 informs about E. coli, S. aureus and phosphatidylcholine liposomes usage in the tests (but not about „phosphatidylcholine liposomes of E. coli and Staphylococcus aureus“ as stated in the manuscript of the minireview)

4. Inhibition of lipopolysaccharide-induced TLR4 signalling and peptidoglycan-induced TLR2 signalling pathways by EGCG has nothing to do with „blocking the formation of bacterial cell walls“; (It must have been a misunderstanding of the message in source publications Nos. 35-37.)

5. The sentences „A study of the effect of EGCG on Hepatitis C virus (HCV) RNA replication, assembly, or release of progeny virions showed no effect. However, it potently inhibited Cell‐culture–derived HCV entry into hepatoma cell lines as well as primary human hepatocytes [41]“ should be rewritten, as they are confusing.

I would appreciate a brief reflection on economic background of a possible usage of TP as antimicrobial agents in plant production (i.e. if their wide usage would be financially acceptable).

Author Response

The text should be revised in order to correct the spelling mistakes. The abbreviated species names of fungi should be corrected (e.g. B. maydis instead of „Bi. maydis“.B. cinerea instead of „Bo. cinerea“)

Response: We are sorry about these unaccustomed Latin abbreviations in our manuscript because that we are worried that so many Latin abbreviations will confuse readers, such as B. maydis and B. cinerea, P. cryptogea and P. syringae. According to your kind suggestion, we have corrected all abbreviated species names we used in manuscript.

The following information, contained in the text of manuscript should be revised according to the source publications:

1. The activity of TP on Pestalotiopsis apiculatus and C. fructicola as a disease agents of Tea-tree (Or „Tea-oil tree“); in the Table 1 (source publication No 15).

Note: The tea-tree products can inhibit agents of leaf blight and anthracose, rather than the tea-tree would suffer of those diseases.

Response: Tea trees (Camellia sinensis) and tea-oil trees (Camellia oleifera) are two species of plants. Tea-oil trees (known as oil-seed camellia or tea oil camellia) but not tea trees are the hosts of Pestalotiopsis apiculatus and Colletotrichum fructicola. In order to avoid confusion, we have added the Latin name of tea at the beginning of the revised manuscript, and changed tea-oil trees to the Latin name in Table 1.

2. The text „A previous study revealed that the electrolyte leakage percentage of rice blast fungus treated with various concentrations of TP changed sharply, indicating that TP treatment changed the membrane permeability of rice.“ (source publication Nr. 14)– probably of rice blast fungus membrane permeability, not the "membrane permeability of rice" itself

Response: We are sorry for our incorrect writing. The statement of “rice” was corrected as “rice blast fungus” in the final version of the manuscript.

3. The source publication No. 31 informs about E. coli, S. aureus and phosphatidylcholine liposomes usage in the tests (but not about „phosphatidylcholine liposomes of E. coli and Staphylococcus aureus“ as stated in the manuscript of the minireview)

Response: We are so sorry for our negligence of this mistake. We have re-read the source publication, and corrected this sentence.

4. Inhibition of lipopolysaccharide-induced TLR4 signalling and peptidoglycan-induced TLR2 signalling pathways by EGCG has nothing to do with „blocking the formation of bacterial cell walls“; (It must have been a misunderstanding of the message in source publications Nos. 35-37.)

Response: Because TLRs (toll-like receptors) are a family of membrane proteins that trigger innate immune responses and are pathogen recognition proteins, we have misunderstood that inhibition of TLR4 signaling and TLR2 signaling pathways could block the formation of bacterial cell walls. We have deleted the statement of “blocking the formation of bacterial cell walls”, and added a speculative sentence in the revised manuscript.

5. The sentences „A study of the effect of EGCG on Hepatitis C virus (HCV) RNA replication, assembly, or release of progeny virions showed no effect. However, it potently inhibited Cell‐culture–derived HCV entry into hepatoma cell lines as well as primary human hepatocytes [41]“ should be rewritten, as they are confusing.

Response: Thank you for your good advice. We have rewritten this sentence according to your comments.

I would appreciate a brief reflection on economic background of a possible usage of TP as antimicrobial agents in plant production (i.e. if their wide usage would be financially acceptable).

Response: Considering your kind suggestion, we have added some descriptions on economic values of a possible usage of TP in Conclusions and Prospects section.

Reviewer 2 Report

The review article entitled "Antimicrobial activities of tea polyphenol on phytopathogens: a review" is a nice approach to compile the importance of Tea polyphenol and their successive use towards development of biological remediation towards growing Phytopathogens.

Though the manuscript could have been written properly to have better understanding. Throughout the manuscript author lacks the standard way of presentation for a nice review article with many grammatical error. Authors are also requested to download some previous review article from the MDPI publishing and read carefully before resubmission to know the proper way of compiling review article.

I strongly recommend to consult a native speaker to improve the article and resubmit. I am pointing some of the minor points:

Line 39: correct the sentence "However, the number of few studies on the..."

Line 61-64: rephrase the sentence.

Line 62: "Of the fungi, Bi. maydis" Authors are requested to follow basic binomial nomenclature to present the genus and species name unless otherwise stated.

Line 65: causea?

Line 65-67: The inhibitive efficiency at the concentrations of 5 mg /mL and 10 mg /mL weres the highest, and the inhibitive ratio achieved 100%.... Check the sentence.

Similarly, antibacterial activity of TPs should be elaborated much in details with clarity and MIC values and discussion.  

Author Response

The review article entitled "Antimicrobial activities of tea polyphenol on phytopathogens: a review" is a nice approach to compile the importance of Tea polyphenol and their successive use towards development of biological remediation towards growing Phytopathogens.

Though the manuscript could have been written properly to have better understanding. Throughout the manuscript author lacks the standard way of presentation for a nice review article with many grammatical error. Authors are also requested to download some previous review article from the MDPI publishing and read carefully before resubmission to know the proper way of compiling review article. I strongly recommend to consult a native speaker to improve the article and resubmit.

Response: Thank you for pointing out the language mistakes in our manuscript. We have invited Bruce J Levine of the University of Maryland College Park for assistance in writing the manuscript.

I am pointing some of the minor points:

Line 39: correct the sentence "However, the number of few studies on the..."

Response: We are sorry for our incorrect writing. We have rewritten this sentence.

Line 61-64: rephrase the sentence.

Response: We have rephrased this sentence.

Line 62: "Of the fungi, Bi. maydis" Authors are requested to follow basic binomial nomenclature to present the genus and species name unless otherwise stated.

Response: Thank you for your good advice. As explained for Reviewer 1, because we are worried that so many Latin abbreviations will confuse readers, such as B. maydis and B. cinerea, P. cryptogea and P. syringae. According to your kind suggestion, we have corrected all abbreviated species names we used in manuscript.

Line 65: causea?

Response: We are sorry for our incorrect writing. The statement of “causea” was corrected as “causes” in the final version of the manuscript.

Line 65-67: The inhibitive efficiency at the concentrations of 5 mg /mL and 10 mg /mL weres the highest, and the inhibitive ratio achieved 100%.... Check the sentence.

Response: We have rephrased this sentence.

Similarly, antibacterial activity of TPs should be elaborated much in details with clarity and MIC values and discussion.

Response: We have added more details about the antibacterial activity of TPs in the final version of the manuscript.

Reviewer 3 Report

The manuscript is well written. However, if the author includes pathways of inhibition mechanism as figures.

It will be good to the acceptable level.

Author Response

The manuscript is well written. However, if the author includes pathways of inhibition mechanism as figures. It will be good to the acceptable level.

Response: We have added a schematic representation of the antimicrobial mechanisms of TP as Figure 2 in the revised manuscript.

Round 2

Reviewer 2 Report

The review article entitled "Antimicrobial activities of tea polyphenol on phytopathogens: a review" is a nice approach to compile the importance of Tea polyphenol and their successive use towards development of biological remediation towards growing Phytopathogens. Although the manuscript has improved a lot after the revision, I still have one point which need to be addressed before acceptance for publication.

In Line 119, 120, 125 and 127 P. s. pv. pisi should be changed to P. syringae or just abbreviate at first appearance under parenthesis like in Line 119: "...against phytopathogenic bacteria (Pseudomonas syringae pv. pisi race 1, P. s. pv. pisi race 2 and P. s. pv. phaseolicola)..." to "...against phytopathogenic bacteria (Pseudomonas syringae pv. pisi race 1 (PS race 1) P. syringae pv. pisi race 2 (PS race 2) and P. syringae pv. phaseolicola PSPP)..." and then represent it as same as abbreviated under parenthesis once appear hereinafter.

Author Response

The review article entitled "Antimicrobial activities of tea polyphenol on phytopathogens: a review" is a nice approach to compile the importance of Tea polyphenol and their successive use towards development of biological remediation towards growing Phytopathogens. Although the manuscript has improved a lot after the revision, I still have one point which need to be addressed before acceptance for publication.

In Line 119, 120, 125 and 127 P. s. pv. pisi should be changed to P. syringae or just abbreviate at first appearance under parenthesis like in Line 119: "...against phytopathogenic bacteria (Pseudomonas syringae pv. pisi race 1, P. s. pv. pisi race 2 and P. s. pv. phaseolicola)..." to "...against phytopathogenic bacteria (Pseudomonas syringae pv. pisi race 1 (PS race 1) P. syringae pv. pisi race 2 (PS race 2) and P. syringae pv. phaseolicola PSPP)..." and then represent it as same as abbreviated under parenthesis once appear hereinafter.

Response: Considering your kind suggestion, we have revised these abbreviates.